# RETQuerySeg: Very-low-parameter adaptation of retinal foundation models for segmentation with query vectors

**Abstract.** Timely segmentation of retinal structures and lesions is critical for screening diseases such as diabetic retinopathy (DR) and retinopathy of prematurity (ROP), yet conventional models remain data- and compute-intensive. We introduce RETQuerySeg, a simple approach for adapting retinal foundation models for image segmentations by learning a single query vector. Concretely, we take the final vector embeddings for each patch, take the dot product with our learnable query and use the result as our predicted segmentation. We apply our approach to two openly available datasets, IDRiD and HVDROPDB, on three segmentation tasks using RETFound-Green as the foundation model. RETQuerySeg achieved strong performance for optic disc segmentation (AUC: 0.9995, Dice: 0.9072) and reasonable performance for ROP ridge segmentation (AUC: 0.9847, Dice: 0.5699), demonstrating generalizability across adult and neonatal retinal images. Performance was limited for small diabetic lesions (AUC: 0.9159, Dice: 0.1173), reflecting the coarse spatial resolution constraint of the patch-based approach. While not achieving pixel-perfect segmentation, RETQuerySeg offers exceptional parameter efficiency, modularity, and computational advantages. Multiple segmentation tasks can be performed simultaneously with minimal additional compute cost, and segmentations can be obtained virtually for free when computing image-level embeddings for classification tasks, making it valuable for resource-constrained settings and explainable AI applications in retinal image analysis.

**Keywords:** Medical Segmentation, Foundation Models, Vector Embeddings.

## 1    Introduction

The retina is a light-sensitive tissue at the back of the eye that allows us to see. Retinal diseases such as diabetic retinopathy (DR) [1] cause vision loss leading to reduced quality of life [2]. Screening for retinal disease with colour fundus retinal images is important as patients might only notice issues with their vision once irreversible damage to their retina has already occurred, whereas early detection allows for sight-preserving treatment. Retinal imaging is also key in detecting and treating retinopathy of prematurity (ROP), which affects low-birthweight premature infants. The burden of diseases like DR and ROP is set to increase globally, especially in developing countries which already face a shortage of specialist ophthalmologists [1], [3], [4], [5]. Deep

learning-based retinal image analysis is a mature field, with promising progress [6], [7], [8], [9], [10], [11], [12], including commercially available models [13], [14]. However, challenges remain particularly around generalisability and robustness [15].

A key challenge for medical AI more generally is the lack of large-scale datasets [16]. So-called foundation models, large deep learning models that have been pre-trained on vast amounts of domain specific data, could address this issue [17], [17], [18]. In ophthalmology, the first foundation model for colour fundus images was RET-Found [19], which spurred a further developments including DERETFound [20] which achieves competitive performance with less pre-training data and RETFound-Green [21]which is even more data efficient while also requiring substantially less compute.

These foundation models are vision transformers [22] which first split an image into patches, tokenize these patches, and then use transformer blocks [23] to process them. Typically, the final representations of each patch are then averaged to obtain a single vector per image for fine-tuning or "linear probing", i.e. adapting the foundation model by fitting a linear model to the final vector embedding.

In this work, we investigate whether retinal foundation models can be adapted for image segmentation and introduce RETQuerySeg. Instead of averaging, we keep the vector embeddings for each patch and learn a simple, 384 parameter query vector to adapt the model for segmentation. We test our approach across three diverse tasks on two datasets – one of adult retinal images, one of neonatal retinal images – and find promising results. RETQuerySeg main benefit is not in providing pixel-perfect segmentations. Instead, it is extremely parameter- and compute-efficient, inherently modular, and allows to obtain segmentations virtually for free if we already use the foundation model for image-level predictions.

## 2        Methods

### 2.1        RETQuerySeg

We propose RETQuerySeg, a method for adapting retinal foundation models for segmentation by learning a query vector $q$. First, we use our pre-trained foundation model $f$ to obtain a vector embedding $v_i$ for each patch of our input image $x$, instead of averaging across all locations as we would when using the model for classification tasks $\boldsymbol{v} = f(\boldsymbol{x})$.

We then take dot product between the query and the vectors for each location, apply an element-wise sigmoid activation, and use that as the prediction for a given patch $p(\boldsymbol{y}|\boldsymbol{x}) = \sigma(\boldsymbol{v} \cdot q)$. We can then fit the parameters of $q$ by minimising the binary cross entropy loss between our predictions and a segmentation label $\boldsymbol{s}$ for all $n$ images in our training set, $\min_{q} \sum_{i}^{n} [-\boldsymbol{y}_i \log \sigma(f(\boldsymbol{x}_i) \cdot q) + (1 - \boldsymbol{y}_i)(\log(1 - \sigma(f(\boldsymbol{x}_i) \cdot q)))]$.

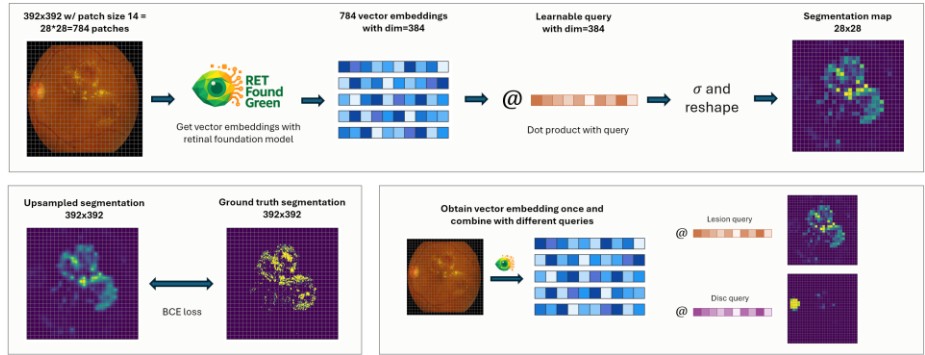

**Fig. 1.** An overview of our proposed RETQuerySeg approach. **Top:** We obtain a vector embedding for each patch of the input image, take the dot product with our learned query vector and apply a sigmoid activation to get a prediction for each patch. **Bottom left:** During training, we use parameter-free bilinear upsampling to compute the binary cross entropy loss at a higher resolution. **Bottom right:** During inference, we only need to compute the vector embeddings once and can then very cheaply obtain segmentations for various target classes.

An overview of our approach is given in Fig. 1. Concretely, RETFound-Green processes images at a resolution of 392x392 pixels and uses a patch size of 14 pixels. 392/14=28, thus we have 28x28 patches. The input resolution and patch size are two key reasons why we use RETFound-Green rather than the original RETFound. The latter uses an input resolution of 224 and a patch size of 16 pixels, thus it only has 14x14 patches in its internal resolution. Another benefit of RETFound-Green is the computational efficiency.

The internal dimension of RETFound-Green is 384, i.e. each vector embedding consists of 384 floating point numbers. Thus, our learnable query vector also has a dimensionality of 384. Taking the dot product between the vector embeddings and each query yields a single number for each patch.

As an implementation detail, we use bilinear upsampling to bring our prediction up to the input resolution so we can compute the loss with the segmentation label. Bilinear upsampling is parameter-free and the foundation model is kept frozen, so we only need to learn the 384 query parameters.

We do not propose RETQuerySeg as a way to achieve optimal segmentations: The relatively coarse spatial resolution of our vector embeddings is of course an obvious limitation of our approach, and indeed we would not expect it to be competitive with standard segmentation approaches like UNets [24]. Instead, our approach has other advantages compared to standard image segmentation.

First, it is a very-low-parameter approach, as we only fit a single 384-dimensional vector. This query vector can then be very easily shared and stored. Second, for inference, virtually all the computational cost is due to obtaining the vector embeddings themselves, while the dot product is a rounding error. Thus, if we have multiple query vectors relating to different segmentation tasks, we can obtain multiple segmentations at essentially no additional compute cost. Third, as we leverage a pre-trained foundation model that is kept frozen, our approach is inherently modular. New query vectors –

either queries trained on one's own data or ones that are shared by others – can easily be added to an inference pipeline. Fourth, when computing average per-image vector embeddings for linear probing, we can again obtain RETQuerySeg's coarse segmentations without meaningful additional compute. These segmentations could then be used for explainability and to highlight possible areas of concern to a clinician.

## 2.2    Datasets

We use two openly available datasets of retinal colour fundus images. First, the Indian Diabetic Retinopathy image Dataset (IDRiD) [25]. IDRiD contains images related to Diabetic Retinopathy (DR), a key retinal disease that is a leading cause of sight-loss worldwide [1], [5]. The dataset has annotations for four types of DR-related lesions (microaneurysm, hemorrhages, hard exudates, soft exudates) which we aggregate into a single label for the present manuscript. IDRiD further has labels for the optic disc, a key anatomical structure in the eye where the nerves and blood vessels pass through the retina. Second, the HVDROPDB dataset [26] of neonatal colour fundus images in the context of retinopathy of prematurity (ROP). ROP is a condition affecting low-birth-weight premature infants and can lead to blindness if untreated. HVDROPDB has segmentation labels for the ROP "ridge", a key landmark which is the boundary between vascularized and unvascularized retina.

IDRiD has an official train-test-split with 54 images for training and 27 images for testing. HVDROPDB has 100 images with ridge labels. Two cameras were used to acquire the images: RetCam (Clarity MSI, US) and Neo (Forus Healthcare, Bangalore, India), with exactly half of the images being from either camera. We randomly select 25% of the images for each label for testing.

## 2.3    Experimental setup

We initialise our query with all zeroes and then train for 150 epochs using a batch size of 16 using the AdamW [27] optimizer with a learning rate of $10^{-2}$ and weight decay of $10^{-8}$ using binary cross entropy as loss. Note that RETFound-Green is kept frozen throughout, we only optimise the parameters of our query vector. The learning rate is linearly warmed up from 0 for the first 10 epochs and then after epoch 50 decayed to $10^{-4}$ using a cosine schedule [28]. For the training set, we use random rotation, flipping, brightness and contrast, as well as scaling as data augmentation. Images are resized to 392x392 and we compute the segmentation loss at that resolution after parameter-free bilinear upsampling of the predictions from 28x28.

As metrics we use the area under the receiver operating characteristic curve (AUC) and Dice score. The AUC captures whether the probabilistic predictions correctly rank positive and negative pixels, while Dice score evaluates performance at a single binarization threshold of 0.5. For AUC, we use an exact, efficient implementation kindly provided by the authors of [29] which allows us to calculate the AUC across millions of individual pixels. We compute the metrics at the image-level and report the average, worst, and best case performance for each metric.

# 3 Results

## 3.1 Optic disc segmentation

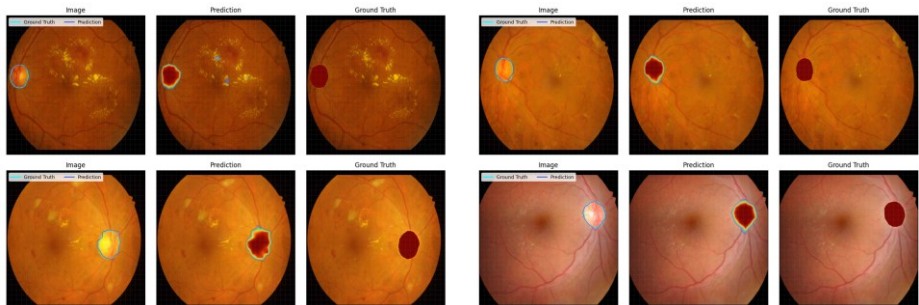

**Fig. 2.** Example optic disc segmentations for held-out test images from IDRiD.

For disc segmentation, RETQuerySeg achieved an average AUC of 0.9995 (min: 0.9988, max: 0.9999) and an average Dice score of 0.9072 (min: 0.8130, max: 0.9634) on the IDRiD test images. Fig. 2 shows some examples. Overall, the disc is correctly identified in all images. However, for the concrete outline, the predictions show some deviation. This is primarily due to the fact that the disc is a round structure, while the patches are square. In some cases, the bilinear interpolation can approximate the outline quite well (e.g. Fig. 2 top left), but in other cases it does so less well (e.g. Fig. 2 bottom left).

## 3.2 Diabetic lesion segmentation

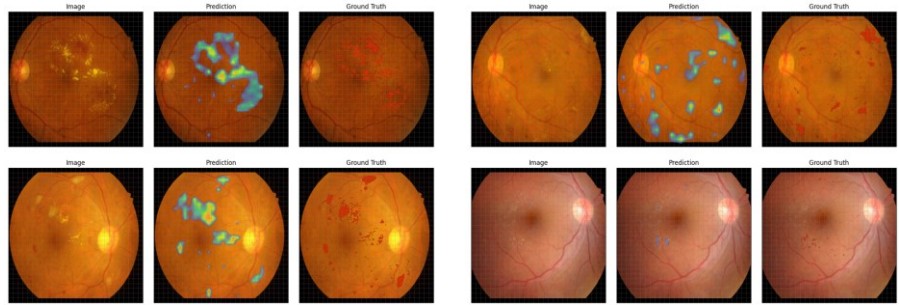

**Fig. 3.** Example lesion segmentations for held-out test images from IDRiD.

Performance for segmentation of diabetic lesions was much worse overall. The an average AUC of 0.9159 (min: 0.7737, max: 0.9685) indicates somewhat reasonable performance, while the average Dice score of 0.1173 (min: 0.0000, max: 0.3916) suggests quite poor performance. This is not entirely unexpected, as some of the diabetic lesions are much smaller than the patch size. In such cases, RETQuerySeg can indicate that a patch likely contains a lesion, but not where exactly the lesion is.

Relatedly, the model rarely has high confidence, unless a whole patch belongs to a lesion. If we changed the binarization from p>0.5 to p>0.1, the average Dice score would improve to 0.3680 (min: 0.1210, max: 0.5779) which is still quite poor, but a meaningful improvement. This indicates that our model is underconfident for lesion segmentation.

However, qualitatively the performance is not as poor as the Dice score suggests. Fig. 3 shows some example segmentations. Generally, the model appears to recognize all larger lesions, areas with many small lesions, and even some isolated smaller lesions. The bottom right image in Fig. 3 is the worst case performance of RETQuerySeg in the test set, which only has comparatively few, very small lesions.

### 3.3    Retinopathy of prematurity ridge segmentation

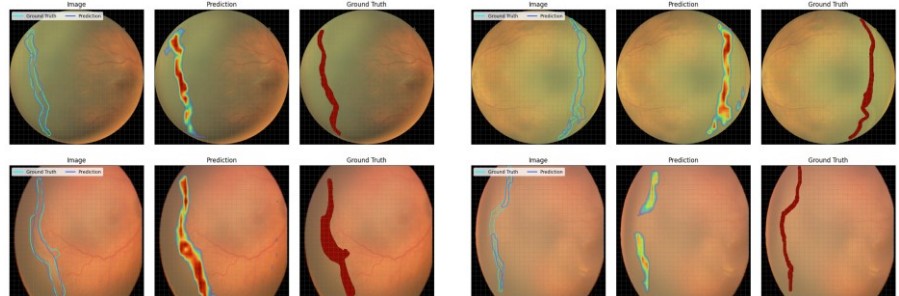

**Fig. 4.** Example ridge segmentations for held-out test images from HVDROPDB.

For segmenting the ridge in ROP images, performance was in-between the previous two extremes with an average AUC of 0.9847 (min: 0.9367, max: 0.9968) and an average Dice score of 0.5699 (min: 0.0824, max: 0.8267). This is in part due to the coarseness of the labels and the challenging nature of taking retinal images of neonatal infants. In many images, the location of the ridge is obvious in parts of the images and speculative in other parts of the images due to blurriness and imperfect illumination.

Fig. 4 shows some example segmentations. While there are clear differences between the labels and predictions, the general location of the ridge in the predictions matches that of the ground truths. Taking a closer look at the ground truths themselves, the width of the ridge appears to be somewhat approximate, i.e. exact pixel-wise agreement is not necessarily desirable, the clinical relevant aspect is the general location of the ridge.

### 3.4    Inference efficiency

We measured the time it took to process a batch of 25 test images on a low-end GPU workstation (Intel i5-12600k, Nvidia 5070ti). We are reporting the mean ± standard deviation across 20 runs. On CPU, obtaining the vector embeddings for all 25 images required $2.7714 \pm 0.0564$ seconds, while on GPU it only took $0.1301 \pm 0.0569$ seconds. Thus, for obtaining the vector embeddings, the CPU was about 21 times slower than

GPU. Once the vector embeddings were computed, taking the dot product with the query and applying the sigmoid activation only took $0.0014 \pm 0.0026$ second on CPU and $0.0001 \pm 0.0001$ on GPU. In other words, computing the vector embeddings themselves takes orders of magnitude more compute, while the RETQuerySeg segmentations are effectively a rounding error for the overall computational requirement.

## 4    Conclusion

Overall, our RETQuerySeg approach yielded reasonable results across three segmentation tasks and two diverse datasets. For the large optic disc, performance was generally quite good, with only small errors at the margins owing to the limitation of an internal resolution of 28x28. Similarly, for ROP ridge segmentation, performance was generally good with errors partially attributable to poor image quality and coarse annotations. However, for diabetic lesions, performance was poor in terms of Dice and only reasonable in terms of AUC, highlighting the limitations of our approach. Especially isolated lesions that are much smaller than the patch size of RETFound-Green were not detected. Still, qualitatively examining the predicted segmentations shows that RETQuerySeg still highlighted the main areas of pathology.

Regarding the ROP results, a further point of note is that RETFound-Green was not pre-trained on any neonatal images and instead had previously only seen retinal images of adults. Neonatal images are quite different in terms of anatomy, pathology, image quality, and use dedicated cameras distinct from those used for adults. Thus, the ROP results demonstrate some generalisability of our approach.

Our approach clearly does not provide optimal segmentation and is not meant to compete with traditional segmentation methods either. Instead, it has a few key advantages that make it useful in settings where other segmentation approaches would not be applicable. RETQuerySeg is not only very parameter efficient and thus the queries are very easy to share, but virtually all inference compute is used for the foundation model itself. This means that we can obtain multiple predictions using different queries without meaningful increase in compute cost. If we are computing image-level vector embeddings anyway for linear probing, we can also obtain RETQuerySeg-based segmentations without additional cost. Finally, RETQuerySeg is modular as the underlying foundation model stays unchanged. Thus, we can add new queries to an existing pipeline.

Beyond more comprehensive experiments, including in other medical imaging modalities, there are a few interesting directions for future work. First, overcoming the limitation of the internal resolution. This could be accomplished with a foundation model pre-trained for a higher resolution or by relaxing the simplicity of our approach, e.g. combining the patch-wise features with pixel-level features (pixel values, basic edge detection, etc.), or fitting a small convolutional decoder instead of a simple query vector. Second, we are interested in using the RETQuerySeg segmentations to take weighted averaged for image-level linear probing. Currently, RETFound-Green uses simple average pooling, which might degrade performance for tasks where the

pathology of interest is only a small part of the overall image. Weighing the average using a learned query related to the task of interest would not meaningfully affect the efficiency of linear probing, but address this challenge. Finally, we currently learn the query vector using segmentation labels. In future work, we want to explore learning it using image-level labels instead, by using the query for a weighted average pooling followed by a linear classification layer which we optimise jointly with our query. This could provide model-level explainability by investigating what features the query is highlighting, and faithful instance-level explainability since we can visualise which areas of an image were weighted heavily for a given prediction.

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
