# OpenReview forum: "RETQuerySeg: Very-low-parameter adaptation of retinal foundation models for segmentation with query vectors"
_MICCAI.org/2025/Workshop/MSB_EMERGE — Submitted to MSB EMERGE 2025_

### Official Review · Reviewer_Toi7 · 2025-07-04

**Recommendation:** 1
**Confidence:** 3

**Clarity:**

The paper is generally clear but has some clarity issues that could be addressed with moderate revision

**Feedback:**

The idea and the experiments seem relevant. Nonetheless, it is unclear what the novelty of this paper is because the related work is missing. Are the authors introducing this vector query idea? Also, no baseline comparison has been done to evaluate how well the method is working against other segmentation methods. If you want to claim the method is efficient, you have to show that it is efficient against those traditional segmentation methods.

Reporting metrics in a Table (for each dataset as well) will make the paper clearer.

**Justification:**

The authors only evaluated the proposed method without performing any comparison. Also, the novelty of this paper is not clear due to the lack of related works discussed in the introduction.

**Reproducibility:**

Sufficient amount of details available for reproducing the main results, but open access is not provided to source code and/or data

**Strengths:**

-	**Method is evaluated for three segmentation tasks:** optic disk, diabetic lesion, retinopathy of prematurity ridge
-	Two public datasets have been used
-	The proposed method is efficient when running on a GPU and can achieve multiple tasks simultaneously

**Summary:**

In this paper, the authors introduce a simple approach, RETQuerySEG, to adapt retinal foundation models to segmentation tasks by learning a query vector and all of the patch embeddings instead of leveraging. They demonstrate that their method works on 3 different tasks and 2 datasets.

**Weaknesses:**

-	**Related work is missing:** The reviewer doesn’t know if the idea is new because the related work is missing. Has this idea already been applied to other foundation models?
-	**No comparison:** Only the proposed method is evaluated. No comparison to any other segmentation methods or against the average embedding. Same comment for the computing efficiency experiments.
-	**Lack of justification** for why we don’t need, as the authors wrote, “pixel-perfect segmentation” for the task

---

### Official Review · Reviewer_yKBP · 2025-07-08

**Recommendation:** 1
**Confidence:** 5

**Clarity:**

The paper is unclear and difficult to understand due to significant clarity issues, major revision is necessary

**Feedback:**

I recommend the authors revise the submission to meet the page limit before resubmitting. Compliance with formatting requirements is essential for fair and consistent review.

**Justification:**

Desk reject due to length policy violation.

**Reproducibility:**

Not applicable: the paper has no experiments

**Strengths:**

Not evaluated due to length policy violation.

**Summary:**

I'm unable to review this submission because it exceeds the maximum length allowed by the MICCAI EMERGE formatting guidelines (longer than 8 pages). Papers that do not comply with submission policies should be rejected without review.

**Weaknesses:**

Length policy violation.

---

### Official Review · Reviewer_Pk6C · 2025-07-09

**Recommendation:** 1
**Confidence:** 5

**Clarity:**

The paper is unclear and difficult to understand due to significant clarity issues, major revision is necessary

**Feedback:**

The paper has major flaws

1. Its unclear what the motivation of the paper is. Is it to improve speed ? Is it to improve accuracy ? What is the problem with current methods ? The paper needs to address these before proposing a solution
2. The paper needs more experiments with current state of the art methods for comparison

**Justification:**

The paper lacks clarity and rigor in experiments.

**Reproducibility:**

Not enough amount of details available for reproducing the main results, and open access details are unclear

**Strengths:**

The proposed method is fast and simple. This provides a sped and compute advantage for screening tasks in practice

**Summary:**

In this paper the authors provide a simple and fast method to segment retinal structures and lesions by segmenting the image into multiple patches and using a query vector to produce a score for the patch. The authors validate their results using IDRiD and HVDROPDB datasets, on three segmentation tasks using RETFound-Green as the foundation model

**Weaknesses:**

1. **Lack of comparison to state of the art** - The paper only provides comparisons to ground truth, but none of the existing methods. The paper does not even provide the baseline results for existing methods.
2. **Paper lacks proper comparisons** - While the authors provide some visual comparisons its very hard to tell if the their approach is better than any existing methods
3. **Lack of clarity** - This paper is hard to read. The numbers of results are mentioned in paragraphs instead of readable tables. While image comparisons are provided, its unclear if this method is good at all.